# Using nasal sprays to prevent respiratory tract infections: a qualitative study of online consumer reviews and primary care patient interviews

Sian Williamson [ID],[1] Laura Dennison [ID],[1] Kate Greenwell [ID],[1] James Denison-Day,[1] Fiona Mowbray,[1] Samantha Richards-Hall,[1] Deb Smith,[1] Katherine Bradbury,[1] Ben Ainsworth,[1,2] Paul Little [ID],[3] Adam W A Geraghty [ID],[3] Lucy Yardley[1,4]

SW and LD are joint first authors.

For numbered affiliations see end of article.

**Correspondence to**
Dr Laura Dennison;
l.k.dennison@soton.ac.uk

## ABSTRACT

**Objectives** Nasal sprays could be a promising approach to preventing respiratory tract infections (RTIs). This study explored lay people's perceptions and experiences of using nasal sprays to prevent RTIs to identify barriers and facilitators to their adoption and continued use.

**Design** Qualitative research. Study 1 thematically analysed online consumer reviews of an RTI prevention nasal spray. Study 2 interviewed patients about their reactions to and experiences of a digital intervention that promotes and supports nasal spray use for RTI prevention (reactively: at 'first signs' of infection and preventatively: following possible/probable exposure to infection). Interview transcripts were analysed using thematic analysis.

**Setting** Primary care, UK.

**Participants** 407 online customer reviews. 13 purposively recruited primary care patients who had experienced recurrent infections and/or had risk factors for severe infections.

**Results** Both studies identified various factors that might influence nasal spray use including: high motivation to avoid RTIs, particularly during the COVID-19 pandemic; fatalistic views about RTIs; beliefs about alternative prevention methods; the importance of personal recommendation; perceived complexity and familiarity of nasal sprays; personal experiences of spray success or failure; tolerable and off-putting side effects; concerns about medicines; and the nose as unpleasant and unhygienic.

**Conclusions** People who suffer disruptive, frequent or severe RTIs or who are vulnerable to RTIs are interested in using a nasal spray for prevention. They also have doubts and concerns and may encounter problems. Some of these may be reduced or eliminated by providing nasal spray users with information and advice that addresses these concerns or helps people overcome difficulties.

## INTRODUCTION

Respiratory tract infections (RTIs) such as the common cold, influenza, bronchitis, tonsillitis and sinusitis are commonly experienced

## STRENGTHS AND LIMITATIONS OF THIS STUDY

⇒ This is the first research about how people think and feel about using a nasal spray to prevent respiratory tract infections so adopting an exploratory, inductive, qualitative approach allowed insight into key issues.

⇒ The paper benefits from its coverage of multiple *populations*, *data collection approaches* and *contexts*.

⇒ The pandemic context, short study period and season meant study 2 participants had little exposure to viruses and limited opportunities to try out their sprays.

⇒ The study 2 sample lacked ethnic diversity, tended to have low levels of deprivation and consisted of more females than men.

⇒ This paper demonstrates the benefit of conducting in-depth qualitative research with target users during intervention planning, development and refinement.

by most adults. Although they tend to be self-limiting, these illnesses are disruptive and unpleasant,[1–5] cause substantial workplace sickness absence[6] and contribute significantly to pressures on primary care.[7 8] Consultations for RTIs also result in unnecessary antibiotics prescriptions, thus contributing to antibiotic resistance.[9 10]

Typical RTI prevention approaches reduce the likelihood of becoming infected (eg, social distancing,[11] face coverings[11] and handwashing[11 12]) or improve individuals' immune responses (eg, vaccination,[13–15] nutrition,[16 17] physical activity[18 19]). Prevention approaches can also intervene at early stages of infection by targeting the nose and the mouth as entry points for viruses.[20] These approaches include using mouthwashes and rinses and nasal sprays, douches and irrigation. Products

may be used regularly and/or in responsible to possible/ probable exposure. The mechanism of action appears to be mechanical; either forming a barrier or having a washing out action. These products may also alter the environment of the nose and/or throat, reducing the viral load and the chance the virus will survive/ thrive.[20 21] The COVID-19 pandemic has prompted a resurgence of interest in these approaches.[20 22–27] Many formulations and products are under investigation, with some promising findings. For example, a systematic review concluded that iota-carrageenan nasal sprays have a good safety profile and powerful antiviral activity against the common cold.[21] A series of recent reviews and commentaries conclude that these approaches should be subject to further evaluation and/or rapid roll-out in the COVID-19 pandemic. Various randomised controlled trials are ongoing. The RECUR (Reducing common infections in usual practice for recurrent respiratory tract infections) trial (ICTRN17936080) evaluates preventative use of nasal sprays to reduce the frequency, duration and severity of non-pandemic RTIs in recurrent and at-risk primary care patients while the ICE-COVID trial[24] evaluates throat and nasal sprays for COVID-19 prevention in healthcare professionals (HCPs).

Along with evidence about efficacy, it is also essential to accrue evidence about the acceptability of these approaches for the people who may eventually be encouraged to adopt them. Kramer and colleagues[20] describe nasal rinsing as 'easily implementable' as a COVID-19 public health measure. However, lay people/patients may not find these approaches easy or acceptable.[28]

No published research has investigated views or experiences of using these approaches for preventing RTIs. However, research exists on similar approaches when used for symptom relief. People with chronic rhinosinusitis describe difficulties using steroid nasal sprays including forgetting to use them, and lack of confidence with technique.[29] It may be considered awkward, prohibitively time consuming[29] and uncomfortable, and, consequently, patients may use these methods irregularly, stopping once relief is gained.[4] Together, these studies indicate that RTI prevention strategies requiring nasal application of a substance may be off-putting for some patients and regular, long-term persistence may be problematic. Identifying concerns and difficulties (along with more positive beliefs and experiences) would allow patient education to be tailored to include persuasive messages and appropriate support to help people overcome barriers.

This paper extends the literature by investigating people's perceptions and experiences of using a nasal spray for preventing RTIs. We report findings from two qualitative studies. The first is an analysis of online customer reviews of an RTI prevention nasal spray. The second study analyses interviews with patients heavily burdened by and/or at higher risk from RTIs about their perceptions and experiences of using a nasal spray for RTI prevention. Our aim for both studies was to explore how people think and feel about using nasal sprays to prevent RTIs and to identify barriers and facilitators to the adoption and continued use of sprays. If sprays prove effective in trials, it is important to have a behavioural evidence base to guide interventions that support optimal use. The findings will be valuable to researchers and clinicians seeking to develop or implement RTI prevention approaches, especially those involving nasal sprays or similar prophylactic products such as nasal and mouth rinses and washes.

## METHODS
### Intervention development context
The studies reported in this paper were undertaken as part of the development of a digital behavioural intervention to encourage and support people to use a nasal spray to prevent RTIs (National Institute for Health Research programme grant RP-PG-0218-20005; 'RECUR'). A randomised controlled trial is currently evaluating the efficacy of the nasal spray intervention; within the trial the brand name of the spray is masked. Therefore, this paper simply refers to it as 'the nasal spray'. As a regulated medical device, the safety of the spray has been established. It is available to purchase in the UK and currently retails under £10. The manufacturer instructions advise use at the first signs of a cold. In the intervention under evaluation, participants are also advised to use the spray at first signs of *any* suspected RTI and also in situations where exposure to RTIs is likely (eg, crowded places, close proximity to infected people).

The intervention development work used the person-based approach,[28] which prioritises in-depth qualitative data collection to explore the views and experiences of potential intervention users, in order to understand the context in which users engage with interventions and behaviour change. Figure 1 shows how the studies reported here were used alongside primary qualitative research,[30] a scoping review, behaviour change theory (protection motivation theory,[31 32] social cognitive theory,[33] necessity concerns framework,[34 35] sense model[36 37]) and stakeholder and patient and public involvement (PPI) to develop and optimise the intervention. The two studies reported here influenced the development of 'guiding principles'[28] (online supplemental material 1) and the articulation of programme theory through a logic model for the intervention[38 39] (online supplemental material 2), then enabled iterative changes to the intervention (online supplemental material 3).

### Study 1: online consumer reviews of the nasal spray
#### Data collection
Four hundred and seven customer reviews of the nasal spray were taken from four large commercial websites (comprising 263, 93, 30 and 21 spray reviews each). The websites were selected based on having a large number of spray reviews. All reviews were included (positive, negative) except those which focused on supplier-based issues (eg, damaged product). We also removed reviews that

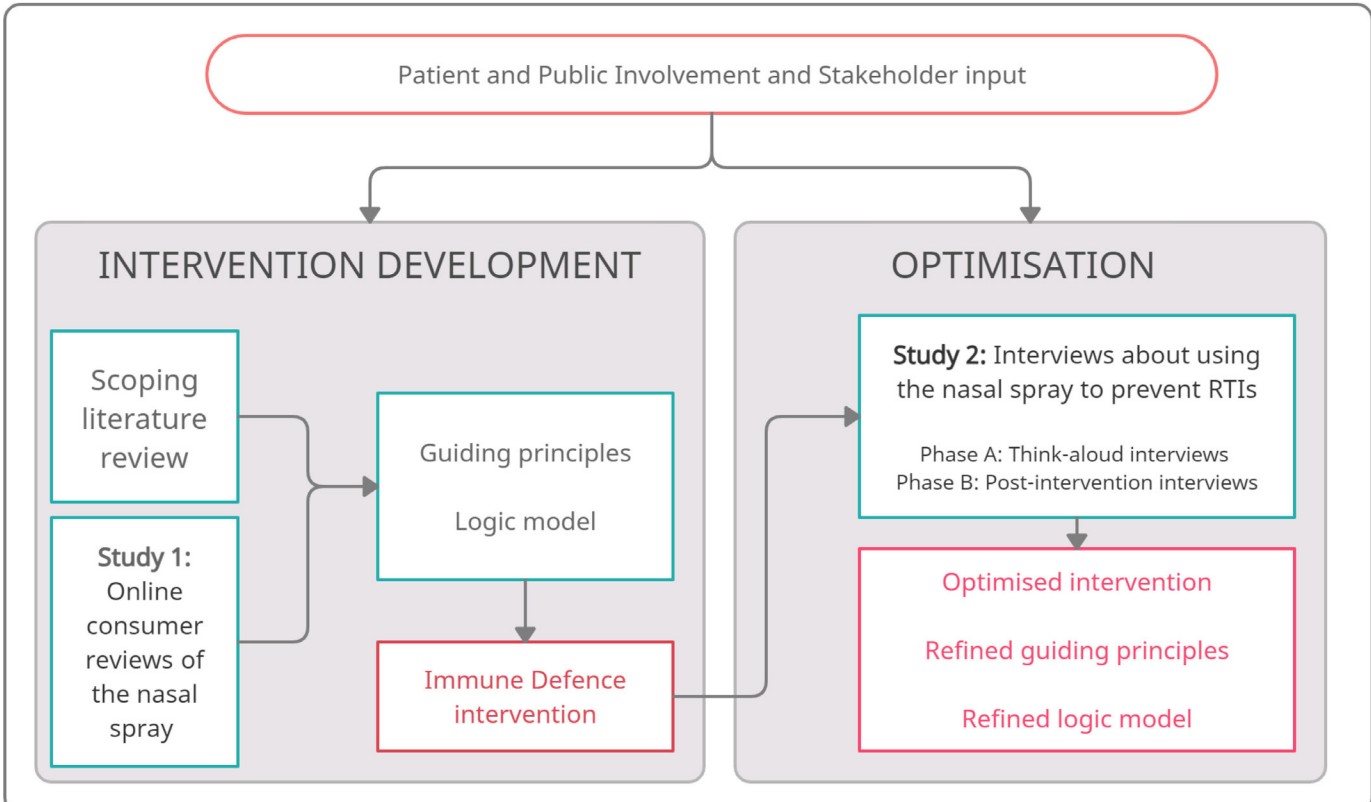

**Figure 1** Overview of nasal spray intervention development activities. RTI, respiratory tract infection.

were duplicated across websites. The search for reviews was conducted in August 2019.

### Analysis

We used an inductive thematic analysis approach. Although the review data were 'thin' and brief (typically several sentences for each review), we selected this approach to remain open and explorative and to generate broad themes that summarised important topics. Coding was undertaken by SW and FM who separately coded half of the reviews each in NVivo V.12 and then worked together to review, combine, discuss and refine coding. They then developed preliminary descriptive themes to capture key issues within the data. These were subsequently inspected, reorganised and relabelled by LD and SW.

### Study 2: interviews about using a nasal spray to prevent RTIs
#### Recruitment

We sought participants who experience frequent or recurrent infections and/or who are at risk of more severe RTIs. Three UK general practitioner practices identified possible participants by searching their patient lists and posting invitations and information sheets to patients who consulted for ≥1 RTI within the last year and were prescribed antibiotics. They also wrote to patients who had asthma, chronic obstructive pulmonary disease or chronic sinusitis who were at higher risk of RTIs. Patients interested in participating returned reply slips, on which they self-reported their recent RTI history. We

then purposively sampled from these responses to prioritise interviewing those with higher RTI frequency and comorbid health conditions. We also sought variation as regards age and gender. We interviewed 13 participants in total.

#### Data collection

Interviews took place from April to August 2020, coinciding with the beginning of COVID-19 pandemic. Consequently, interviews were conducted by telephone. Participants provided written consent prior to taking part. Before the interview, participants answered brief questions about demographics and the number and type of RTIs they experienced.

#### Phase A: think aloud interviews (n=10)

Participants were emailed a link to our prototype web-based intervention promoting nasal spray use for RTI prevention (figure 2 provides an overview of this intervention). They worked through the website while simultaneously sharing their reactions aloud. The researcher prompted them to verbalise their thoughts and feelings as they encountered different pages, sections, messages, images and features.

#### Phase B: postintervention interviews (n=7)

Participants were emailed a link to the digital intervention (now optimised based on phase A feedback). A nasal spray was posted to them along with a short booklet summarising spray instructions. They were asked to use

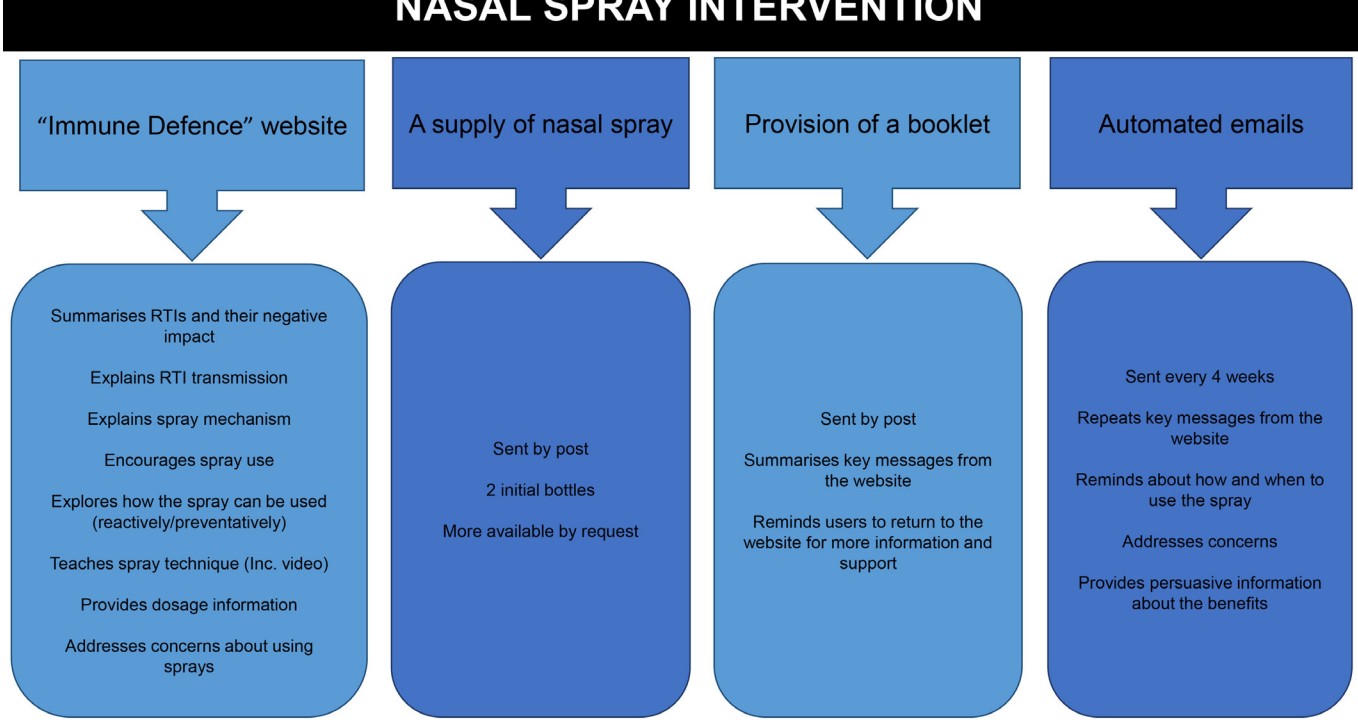

**Figure 2** Overview of nasal spray intervention. RTI, respiratory tract infection.

the website and the spray independently over a period of 2–3 weeks. They then participated in an in-depth interview about their experiences. All participants also answered open-ended questions about their personal experiences of RTIs; findings from this part of the interview are published elsewhere.[30]

Online supplemental material 4 contains the interview schedules. SW and LD conducted the interviews; both are female postdoctoral researchers with health psychology and qualitative interviewing expertise. Interviews lasted between 46 and 104 min and were audio recorded and transcribed verbatim with identifying details removed. Participants received a £10 voucher to thank them for their time.

## Analysis
We used an inductive thematic analysis approach. Transcripts from phases A and B were analysed together. The analysts familiarised themselves with the audio recordings and transcripts. Line-by-line coding of the data was conducted in NVivo V.12 whereby codes were identified and labelled to capture references to perceptions or experiences of nasal sprays for preventing RTIs. The codes were then reviewed, compared, discussed and progressively clustered and merged in order to create themes. This was an iterative process which progressed to refining and organising final themes that captured important patterns and features in the data. SW and LD led the analysis, and all other authors were involved in interpreting, discussing and finalising themes. The research team have health psychology and medical backgrounds and the lead analysts are experienced qualitative researchers.

## Patient and public involvement
A panel of PPI contributors with experience of recurrent RTIs and/or health conditions that mean they are vulnerable to frequent or severe infections have inputted into the study planning and conduct, some from the grant application stage. Contributions included editing and improving our participant information sheets, consent forms and interview schedules and participating in pilot interviews helping to interpret findings and drafting this paper and lay summary of the research findings sent to participants. Two members of the PPI panel are coauthors on this paper (DS and SR-H).

This research has been reported in line with the Consolidated Criteria for Reporting Qualitative Research checklist (online supplemental material 5).

## FINDINGS
### Study 1: online consumer reviews of the nasal spray
Eight themes about nasal spray experiences were developed from the customer review data. These are described below and supporting quotations are provided in table 1. The wording of illustrative quotations has been edited slightly to prevent the original reviews and reviewers being identifiable (eg, through entering the quotation into a search engine). SW reworded the quotations, keeping meaning as close to the original as possible. LD checked and further edited reworded quotes to ensure it retained the meaning and could not be traced back to the original review.

**Table 1** Themes from study 1

| Theme | Illustrative quotations |
|---|---|
| Motivation to avoid infections | 'As a mum, I can't afford to be ill – so it's wonderful that I now don't even though the rest of the family do.'<br>'Because of my COPD I have to be careful cos colds can turn into a chest infection.' |
| Inevitability of infections | 'In my opinion, when you've got a cold there is no way to stop it.' |
| Alternative approaches to infection prevention | 'My body would probably have got rid of the cold—it usually does with vitamin c, drinking honey and using a salt water spray for my nose.'<br>'In my opinion, if you don't touch your face (mouth, eyes and nose), this will prevent a cold. Germs live on surfaces for hours, so we need to be aware of this when we are out and about but especially if any of our family have an infection.' |
| Recommendations from others | 'I bought the spray because a nurse recommended it.'<br>'My husband is a strong believer in this stuff.' |
| Protection from risky situations | 'I use it for the Tube where lots of people might be unwell—sneezing and stuff. The spray says to use it for when you have a cold coming but I have been using it every day regardless.'<br>'I purchased it for when I go on holiday, when I usually catch infections when travelling by airplane. Since using it, I've not had any colds on my last two trips.' |
| Ease of spray use | 'The spray is easy to use and you can take it anywhere with you. I don't go anywhere without it.'<br>'The instructions say you should aim towards your ear, and I thought I did do that. It's difficult to do it right.'<br>'If you don't catch you first signs really early (eg, the first odd feeling like tickling in the back of your throat) it will be too late. If your nose is already stuffy, it probably won't work.'<br>'You must use the spray for a couple of days after your symptoms have gone away. If you stop when your symptoms are improving, your infection comes back.' |
| Experiencing side effects | 'The negative part is throat pain for 5 minutes or so, but that's the only negative. It's really bad pain but it's worth it to avoid getting a cold.'<br>'I had extreme side effects. I don't want to have them again so I got rid of it. I reckon it works but the side effects were too bad for me!' |
| Expectations and experiences of success and failure | 'Since the start of the year, I'd been unwell all the time. Then I used the spray at first signs and it stopped my cold (or at least made it tolerable and easier to deal with).'<br>'I've used the spray before and believed it had stopped my colds. However, it failed this time even though I followed the instructions exactly! The cold was the worst I've had in ages so now I just don't know if the spray DID work when I used it before.'<br>'There's no way to be sure if my infection would have continued to get worse without the spray but, if there's any chance it was crucial in stopping the cold, then it's worth it!' |

COPD, chronic obstructive pulmonary disease.

### Motivation to avoid infections
Reviewers described strong motivations to avoid becoming ill with cold-like illnesses. For some this was to avoid disruption to responsibilities and routines. Others were focused on avoiding unpleasant or severe symptoms or health complications for themselves or others that they might infect (eg, vulnerable family members).

### Inevitability of infections
Some reviewers were fatalistic about catching colds and similar infections and believed that symptoms would inevitably develop and progress despite using the spray.

### Alternative approaches to infection prevention
Some reviewers described alternative, competing or perceived superior approaches to avoiding RTIs. This included measures such as good hand hygiene, healthy eating and vitamin supplements. Some expressed a perceived confidence in the body's own ability to fight off infections.

### Recommendations from others
Reviewers sometimes described being influenced to buy and try the spray because of success stories and recommendations from others such as friends, family or HCPs.

### Protection from risky situations
Some reviewers described adapting the way that the spray was used, beyond first signs and symptoms of an infection (ie, recommended use as advised on product instructions). They adopted it as a preventative measure for when they perceived a high threat of infection, for example, when travelling or in busy public places.

### Ease of spray use
Reviewers often described sprays as quick and convenient to use and easily incorporated into daily life. However,

**Table 2** Demographic and clinical characteristics of study 2 participants (n=13)

| Characteristic | Summary statistics |
|---|---|
| Type of interview participation, n (%) | |
| Think aloud interview only | 8 (61.54) |
| Postintervention interview only | 3 (23.08) |
| Both think aloud and postintervention | 2 (15.38) |
| Age (years), mean (SD), range | 54.34 (22.24), 18–83 |
| Gender, n (%) | |
| Men | 3 (23.1) |
| Women | 10 (76.9) |
| Marital status, n (%) | |
| Married or living with partner | 5 (38.46) |
| Single | 3 (23.08) |
| Divorced | 2 (15.38) |
| Widowed | 3 (23.08) |
| Employment status, n (%) | |
| In paid work (full time or part time, employed, self-employed) | 4 (30.77) |
| Retired | 4 (30.77) |
| Not working because of illness/disability | 2 (15.38) |
| Other (unemployed, home maker, student) | 3 (23.08) |
| Education (age left education), n (%) | |
| 16 or before | 2 (15.38) |
| 17 or 18 | 3 (23.08) |
| Over 18 | 8 (61.54) |
| Deprivation (IMD*), median (IQR), range | 10 (6.0), 3–10 |
| Ethnicity, n (%) | |
| White British | 7 (53.85) |
| White Irish | 1 (7.69) |
| Mixed—White British/Asian | 1 (7.69) |
| Not provided | 4 (30.77) |
| Health conditions, n (%)† | |
| Asthma | 6 (46.15) |
| COPD | 2 (15.38) |
| Chronic sinusitis | 1 (7.69) |
| None of these conditions | 7 (53.85) |
| Number of RTIs in the last 12 months, mean (SD), range | 2.92 (1.38), 1–5 |
| RTIs per year in the last 3 years, n (%) | |
| ≥1 | 12 (92.31) |
| ≥3 | 7 (53.85) |
| Types of RTIs experienced at least once in the last 12 months, n (%) | |
| Cold | 10 (76.92) |
| Influenza | 2 (15.38) |

Continued

**Table 2** Continued

| Characteristic | Summary statistics |
|---|---|
| Throat infection | 9 (69.23) |
| Chest infection | 7 (53.85) |
| Sinus infection | 6 (46.15) |
| Ear infection | 3 (23.08) |

*IMD=2019 Index of Multiple Deprivation decile, derived from participant postcodes, 1 is the highest deprivation, 10 is the lowest deprivation.
†The percentage totals more than 100 because 2 participants (15.38%) had more than one of these conditions.
COPD, chronic obstructive pulmonary disease; RTI, respiratory tract infection.

some drew attention to the importance for correct technique and timely usage for efficacy. Some found that this is not always easily achievable.

### Experiencing side effects
Reviewers commonly reported side effects including an unpleasant taste or feel in throat or nose, sinus pain, headache or watery eyes. Side effects differed in severity across reviewers. When describing side effects, reviewers often referred to weighing up the experience of side effects against the impact of having a cold-like infection, reaching a range of conclusions about which was most desirable.

### Expectations and experiences of success and failure
Some reviewers expressed confidence in the efficacy of the spray and referred to its ability to completely prevent colds and influenza from developing or at least reduce the severity of symptoms and shorten their duration. Some reported lack of success or inconsistent results whereby sometimes infections happened despite use (although sometimes these were perceived as possibly milder than they would have otherwise been). Some reviewers emphasised the difficulties in judging whether the spray worked or not, given that it was uncertain how symptoms would have developed over time without spray use. However, doubts and uncertainties did not necessarily deter future use.

### Study 2: interviews about using a nasal spray to prevent RTIs
#### Participants
Table 2 describes the study 2 participant characteristics.

#### Themes
Eight themes were developed (table 3). These are described below.

#### Excitement and optimism about a novel prevention method
Overall, participants described positive and optimistic views about using the spray, seeing it as novel and of interest and personal relevance. For a few participants,

**Table 3** Themes from study 2

| Theme | Illustrative quotations |
|---|---|
| Excitement and optimism about a novel prevention method | 'Then, when this came along it was like lightbulbs going off. I'm thinking, oh my God, this is going to be a way that I can safeguard myself and continue to be active within his life. I'm really excited about the uses of it.' (Participant 10)<br>'I would quite happily give it a go.' (Participant 11)<br>'A hundred per cent I'd be up for giving it a go.' (Participant 5)<br>'I will give it a go I can tell you that now.' (Participant 6) |
| Identifying first signs of infection | 'I tend to just feel more rundown, tired, a bit headachy.' (Participant 7)<br>'A lot of the times when I'm sneezing it's just because of my hay fever. It was quite difficult to tell.' (Participant 9)<br>'If I know it's coming, by the time I'm doing something about it, I guess because my immune system's got no great strength, it's almost like too little, too late.' (Participant 10) |
| Considering use in risky situations | 'I can say, "Well, I've got to go out. There's a chance I may be in contact with other people, so I'll use the spray." It's like another layer of protection.' (Participant 12)<br>'COVID-19 makes it more appealing, actually. I was quite intrigued about whether it would work for COVID.' (Participant 11)<br>'I don't know, on a bus, supermarket, at the cinema, at the theatre… Like when you get home from the theatre you could start using it then, even if you haven't had any signs of anything. That was something that I thought was really useful to have. I could see that scenario.' (Participant 10) |
| Consequences of feeling protected | '…that could only be helpful. I'm genuinely interested from those points of view, because I could get protection in the small part of my life where I'm allowing myself to be at risk, plus I think if I felt safer, I might therefore go out more and feel less frightened about the world.' (Participant 10)<br>'It just meant that I could get on with things. Did I feel invincible? No, but I felt like I didn't have to worry too much, whereas I think if I was coming down with a cold I would have worried about work and being ill and not being able to complete work. I felt more relaxed, maybe, more confident.' (Participant 11)<br>'But then would it encourage more people to actually go out and be slightly more reckless with sprays and masks and protection, washing their hands, touching their face because they're going, "Oh, I'm using the spray, it's okay." That's the other side of it.' (Participant 12) |
| Concerns about medicines | 'Part of it is because I don't like using medications, and I particularly don't like nasal sprays. I think over the last year or so I've used far too many and I'm a bit fed up of putting things in my nose. I think there's something off-putting about that.' (Participant 11)<br>'I mean, to be fair, if it worked and it stopped me taking my medication, I'd much rather use a spray than medicine.' (Participant 8)<br>'At the same time, I was like, oh, well if you don't have to ask a doctor and it's not an actual medication is it actually going to work?' (Participant 9) |
| Unpleasantness and hygiene | 'It's not particularly pleasant, is it, watching people sticking things up their noses and their noses run and stuff.' (Participant 11)<br>'You spray it up and then it all runs down. That sounds disgusting.' (Participant 4)<br>'I was also worried that if I used it, it would pour everywhere. It didn't really.' (Participant 9)<br>'I wouldn't [re]use anything that went into an orifice like an inhaler, or something I stuck up my nose, I wouldn't keep it and use it for another time.' (Participant 10) |
| Familiarity and confidence | 'I'm not very good at stuff like that. …I don't think I've ever really tried one [a nasal spray]. I'm just kind of wary of it.' (Participant 9)<br>'It's common sense really. I've been using a [different type of] spray for years.' (Participant 4)<br>'It's so straightforward using a nasal spray… I wouldn't bother with the video… Particularly at my age range, you've probably used nasal sprays several if not many times over your lifetime so you just would just use it.' (Participant 1)<br>'I think [I was] probably arrogant, I probably thought, "Oh, for goodness sake, I don't need to be shown how to use a nasal spray!" Although, clearly I did because once I used it as recommended, I didn't get a headache.' (Participant 13) |
| Reactions to possible or actual side effects | 'I think it's good that it's listing the side effects, but they're not severe side effects. Obviously, if they're only very, very small, like causing a headache, you can take some paracetamol for that. If it stops you getting an infection or prolonging the infection, then a headache, just stopping that is very minor.' (Participant 5)<br>'I'd rather have that then a full-blown infection. That is on the plus side, even if it can cause a headache.' (Participant 8)<br>'I thought I'd try it again, and I did aim it more towards the ear, and although I did get a slight headache, it was much better and it went away very quickly.' (Participant 13) |

there was a very pronounced excitement, with the spray seen as a way of transforming their quality of life. Others were more muted in their enthusiasm but still interested and willing to try the spray. Even participants who were not fully convinced that the spray would work still considered it worth a try.

Participants found the explanations in the Immune Defence digital intervention about how the spray works to be understandable and plausible, in particular how the spray created an inhospitable environment for viruses. These ideas were particularly relevant and persuasive based on understandings about viruses and infection that participants were rapidly developing during the COVID-19 pandemic.

### Identifying first signs of infection

Most participants were aware of their early RTI signs and felt able to recognise and react promptly to them by using a spray. However, sometimes participants found it difficult to tell whether a symptom signalled an oncoming infection. The crossover between hay fever and RTI symptoms was a particular area of uncertainty.

A minority of participants also described how they never experienced common early signs of infection and

only became aware of oncoming illness through a severe symptom typical of a later stage of an infection (eg, cough). Some therefore anticipated struggling to intervene in time.

### Considering use in risky situations

Participants were particularly interested in using the spray in risky situations to prevent infections. Some participants considered that this mode of use may help protect against COVID-19, although some remained cautious.

Some participants easily identified risky situations, where they would be happy to use the spray preventatively such as supermarkets, hospital appointments, concerts, airplanes and public transport. However, other participants debated or expressed uncertainty about what level of exposure would count as 'risky'. For some, most situations were currently considered risky (ie, during the COVID-19 pandemic). Others felt that if other mitigations were in place (such as social distancing or face masks) the spray was redundant for RTI prevention.

### Consequences of feeling protected

A few participants anticipated that the protection against RTIs afforded by the spray would change how they felt, thought and behaved including feeling safer, less fearful, more relaxed and more comfortable mixing with people with RTIs. A minority expressed concern that using the spray could lead to negative consequences for infection prevention behaviours. They speculated that other people (not themselves) might adopt less cautious behaviour overall. This concern appeared to be heightened by the COVID-19 context and included worries that, if other people were using the spray, they might be less likely to engage in other preventative behaviours such as masks and social distancing.

### Concerns about medicines

Participants appeared to perceive RTI prevention nasal sprays as a form of medicine (the spray is officially a 'medical device'). Conceptualisation of the spray in this way seemed to persist for most participants to some degree despite encountering and understanding our intervention message that the spray is not a medicine and our comparison of regular spray use to regular hand sanitising. In line with perceiving the spray as a form of medicine, participants raised questions and concerns that are typical of medicines. For example, they were interested in ingredients and wanted to check for allergies, interactions or contraindications with their routine medications. Participants also expressed apprehensions regarding overuse which they felt could lead to side effects, addiction or the spray becoming ineffective.

Participants often discussed trying to avoid using medicines. While this could raise concerns about using the spray, a few considered the spray a means of avoiding needing medication for RTI symptoms or disease exacerbations (eg, antibiotics, steroids).

Although thinking of the spray as a medicine elicited concerns relating to medicines, thinking of the spray as something without medicine 'status' also appeared problematic; a minority of participants expressed slight doubt about how effective the spray could be if it was not a medicine, and not already regularly prescribed or recommended by the National Health Service.

### Unpleasantness and hygiene

A few participants described how actions relating to noses and nasal mucous were unpleasant and socially unacceptable. A few (specifically those unfamiliar with using any type of nasal spray) found that the concept of a nasal spray inactivating and cleaning out viruses raised concerns about a messy and wet procedure. However, those who tried out the spray did not find this occurred. Given their awareness that viruses might be present in the nose, some participants were also concerned about how to use the spray hygienically. For example, they wondered whether germs left on the nozzle could infect them if they used the spray again later.

### Familiarity and confidence

There was considerable variability in how much detailed information people felt they needed about exactly how to use the spray. This seemed to relate to lack of confidence and was prominent in participants who had not used any type of nasal spray before. One participant found using a spray daunting, was anxious about getting it right and found detailed instructions reassuring. Conversely, participants who had previously used another type of nasal spray appeared comfortable trying a spray and had fewer questions and concerns, seeing it as obvious and commonsense. However, this confidence could be unhelpful; one confident participant bypassed the instructions, tried the spray using the incorrect technique and experienced strong side effects. They described having thoughts about never using the spray again before realising the value of the technique instructions. Generally, people welcomed access to detailed guidance about spray technique and especially appreciated that the tips were aimed at helping them to reduce chance of side effects.

### Reactions to possible or actual side effects

Participants considered knowing about the potential side effects of the spray important, paid keen attention to this information, but overall did not consider them off-putting. Participants stated that they would be willing to try the spray and would review their position and stop using the spray if bad side effects were experienced.

## DISCUSSION

This paper is the first published research to explore how people think and feel about using nasal sprays, an emerging area of RTI prevention. Various important perceptions and experiences were identified which are discussed below in terms of their relevance for

encouraging people to adopt and persist with this type of RTI prevention approach, if trial evidence supports their effectiveness.

### Existing theory and research

Our findings align well with expectancy value theories of health behaviour such as health belief model[40] and the necessity concerns framework.[34 35] These theories emphasise implicit cost-benefit analysis; a person adopts and perseveres with preventative health behaviours generally or adherence to a medicine specifically based on perceived efficacy, necessity and tolerability. We found strong beliefs about necessity in both studies. Study 1 participants wanted to avoid the physical and social impacts of RTIs and study 2 participants (with recurrent RTIs or vulnerabilities to severe RTIs) welcomed our information and advice and considered sprays a novel and potentially effective prevention method. Considerable interest in strategies to prevent RTIs has been recently documented in vulnerable and/or recurrent patients[30] but research with younger and/or healthy participants in non-pandemic times reveals weaker or mixed beliefs about the necessity of avoiding infections.[1 2 41–45] Both studies reported here also highlighted a range of beliefs and concerns that could plausibly reduce engagement with using nasal sprays. Concerns around discomfort and regime complexity also arose in studies about nasal irrigation and sprays for sinusitis relief.[4 29] According to expectancy value theories, reducing concerns and costs (alongside increasing necessity beliefs) will improve initiation and continuation of the behaviour.

A theoretical review[46] argues that medication adherence should be conceptualised as a type of causal learning and reasoning. People learn about how medications effect outcomes through a dynamic interplay of top-down (pre-existing beliefs and expectations about treatments) and bottom-up processes (personal experiences with symptom change and side effects, particularly early in the course of treatment). This learning influences their ongoing adherence. Causal learning theory[46] predicts that learning a link between an intervention and positive outcomes (and therefore strong adherence) in the context of a nasal spray for RTI prevention could be challenging for several reasons. First, people have limited data on which to reach conclusions from (eg, several infections per year, rather than daily symptoms). Second, other variables confound interpretations of spray efficacy (eg, other RTI prevention behaviours). Third, sprays may not prevent infections 100% of the time, especially when use is suboptimal (timing, technique, dosage). Our findings about optimism about the spray are positive; people are likely to begin using sprays with expectancies that will facilitate interpreting a link between the spray and positive outcomes. However, some participants described doubt about effectiveness and some highlighted the difficulty of drawing strong conclusions from one's own experience. This, alongside the identified focus on side effects and concerns about using medicines, suggests that causal

learning of a treatment benefit may be difficult and this may undermine adherence.

Finally, perceived ease or difficulty of using the spray and confidence for using it were also prominent within our findings. Social cognitive theory highlights self-efficacy as a key predictor of behaviour.[47] Intervention complexity and lack of confidence, alongside poor adherence, have also been emphasised in research on nasal irrigation for sinus symptom relief.[4 29]

### Intervention development

We undertook the two studies reported here while developing the Immune Defence nasal spray intervention. Study findings informed the planning of initial intervention content (study 1) and optimisation of that content (study 2). For instance, our intervention content addressed concerns about overusing medicines, side effects and hygiene as well as avoided disgust reactions. We provided persuasive information to challenge fatalism about catching RTIs, helped people to build positive expectations of the spray and to continue to hold these even if it does not appear to work every time. We promoted the benefits of feeling protected, while explaining the importance of continuing other RTI prevention behaviours. We emphasised the simplicity of spray use (and ensured a straightforward experience via clear, easy instructions) and we presented information to suit both experienced nasal spray users and less confident beginners. Online supplemental material 6 provides further details about how study findings influenced intervention content.

### Strengths, limitations and future research

A key strength of this paper was its combination of findings from different samples and data collection methods allowing insights into a variety of people and experiences. Some of our data reflect experiences of people who were already motivated to buy the spray and who had some experience of using it, but we also gathered data from people for whom RTI prevention is clinically relevant but who did not currently use nasal sprays. We also collected data from pre-COVID-19 and early pandemic contexts.

Study 1 was a large sample but collected and analysed thin, brief data with little contextual information and no knowledge of reviewer demographic and clinical characteristics. Furthermore, the reviews cannot be verified as genuine as they were on commercial websites. However, the details of problems, concerns and doubts that were largely supported (and extended) in study 2 give confidence that we have captured genuine data.

Study 2 examined the reactions to the Immune Defence intervention content allowing insight into what is interesting, confusing, concerning, off-putting about the nasal spray as described by a *specific rationale and set of instructions*. While some of the detail is therefore particularly pertinent to the Immune Defence nasal spray intervention, the overall themes may be generalisable to other nasal sprays and similar products, prevention behaviours, instructions and advice. Phase b of study 2 was designed

to explore how people experience beginning to use the spray for the first time. A significant limitation, however, is that only seven participants took part in this phase. They also tried the spray over just 3 weeks, in a partial COVID-19 national lockdown and during the summer months. They therefore experienced little exposure to viruses and consequently had limited opportunity to use the spray in the intended ways. Tracking more participants over longer periods will provide a clearer picture of usage and adherence and will be particularly useful for shedding light on factors that may only become apparent over time (eg, experiencing or not experiencing benefits). Qualitative and quantitative data collection on spray adherence, experiences and beliefs is currently in progress as part of the Immune Defence process evaluation.

While our findings suggest nasal sprays for RTI prevention are of interest to clinically higher risk subgroups and considered particularly valuable in the pandemic context, whether lower risk groups (eg, healthy adults) have similar perceptions has not been established. Furthermore, some of the recent and current trials of nasal sprays and similar approaches relate specifically to HCPs at risk during provision of medical care.[24] Findings about lay people's motivations, facilitators and barriers may not transfer well to HCPs; their expertise and the occupational setting may mean different factors are important. Additional research may therefore be needed with these groups.

## CONCLUSION

People who suffer frequent or severe infections or who are clinically vulnerable to RTIs are interested in using a nasal spray to prevent RTIs and see this as useful or even a 'game changer'. They also have some doubts and concerns and may expect to encounter (or actually encounter) certain difficulties. Many of the information needs, misunderstandings, concerns and difficulties exposed through the current research may be remedied by ensuring interventions are designed to help people overcome these issues.

**Author affiliations**
¹Centre for Clinical and Community Applications of Health Psychology, School of Psychology, Faculty of Environmental and Life Sciences, University of Southampton, Southampton, UK
²Department of Psychology, University of Bath, Bath, UK
³Primary Care and Population Science, University of Southampton, Southampton, UK
⁴School of Health Sciences, University of Bristol, Bristol, UK

**Acknowledgements** Kate Martinson managed the ethical approvals and recruitment. Thank you also to all our PPI panel members, in particular Hazel Patel, SR-H and DS.

**Contributors** LY, AWAG and PL conceived the study idea and initial study design with later input from BA, LD, SW, KG, FM, JD-D, KB, SR-H and DS. SW led the data collection with assistance from LD. SW, LD and FM led the data analysis with input from all authors at different stages. LD and SW drafted the manuscript. All authors contributed to critically editing and approving the final manuscript. AWAG is the guarantor for this work.

**Funding** This study/project is funded by the National Institute for Health Research (NIHR) Programme Grants for Applied Research (PGfAR) programme. This study was nested within an NIHR Programme Grant for Applied Research: REducing Common infections in Usual practice for Recurrent Respiratory tract Infections (RECUR) (PL, AWAG) (RP-PG-0218-20005). LY is an NIHR senior investigator and her research programme is partly supported by the NIHR Applied Research Collaboration (ARC)-West, the NIHR Health Protection Research Unit (HPRU) for Behavioural Science and Evaluation and the NIHR Southampton Biomedical Research Centre (BRC). The research programmes of LY and JD-D are partly supported by the NIHR BRC. The intervention development methods used for the RECUR/'Immune Defence' intervention were developed with support from the NIHR BRC.

**Disclaimer** The views expressed are those of the author(s) and not necessarily those of the NIHR or the Department of Health and Social Care.

**Competing interests** None declared.

**Patient and public involvement** Patients and/or the public were involved in the design, or conduct, or reporting, or dissemination plans of this research. Refer to the Methods section for further details.

**Patient consent for publication** Not required.

**Ethics approval** This study involves human participants and ethics and research governance approvals were granted by the University of Southampton for study 1 (ERGOII:52394). Ethics approvals were granted by the NHS and the University of Southampton review boards for study 2 (REC/HRA19/SC/0354; ERGO:48223). Participants gave informed consent to participate in the study before taking part.

**Provenance and peer review** Not commissioned; externally peer reviewed.

**Data availability statement** No data are available.

**ORCID iDs**
Sian Williamson http://orcid.org/0000-0001-5448-3499
Laura Dennison http://orcid.org/0000-0003-0122-6610
Kate Greenwell http://orcid.org/0000-0002-3662-1488
Paul Little http://orcid.org/0000-0003-3664-1873
Adam W A Geraghty http://orcid.org/0000-0001-7984-8351

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
