## [Reviewer comments · BMJ Open]

ARTICLE DETAILS

TITLE (PROVISIONAL)	Using nasal sprays to prevent respiratory tract infections: A QUALITATIVE STUDY OF ONLINE CONSUMER REVIEWS AND PRIMARY CARE PATIENT INTERVIEWS
AUTHORS	Williamson, Sian; Dennison, Laura; Greenwell, Kate; Denison-Day, James; Mowbray, Fiona; Richards-Hall, Samantha; Smith, Deb; Bradbury, Katherine; Ainsworth, Ben; Little, Paul; Geraghty, Adam; Yardley, Lucy

VERSION 1 – REVIEW

REVIEWER	Mark Ebell University of Georgia, Epidemiology and Biostatistics
REVIEW RETURNED	04-Feb-2022

GENERAL COMMENTS	I'm afraid that I am not a qualitative researcher and therefore can really only comment on the medical/scientific content and the presentation/writing. My review is therefore more limited than usual. While very well written, I found the 5200 word length to be a bit overwhelming. For example, the Introduction could be halved, easily. We know respiratory infections are common and important, don't need as much of that. We don't need an overview of prevention – just tell us about what is known about the efficacy of nasal sprays for prevention. Using a nasal spray to prevent RTI is a new concept to me – is there any evidence of efficacy, or just ongoing trials? The authors seem to make the assumption that it is highly effective, safe and cheap, and we just need to convince everyone to use it. Page 7: Can you at least tell us the ingredients of the spray? I'm not interested in reading further unless I can be convinced it is likely or even possibly effective. Page 8: Approach of using reviews from commercial sites is novel and interesting. How do you know that reviewers actually used the product, though? Table 3 and elsewhere (e.g. page 20, lines 6-14 and below in 'Considering use in risky situations'): I am concerned that from the comments, some users feel as if this is a "magic bullet" and they don't have to take other precautions. This could lead to an increased risk of infection and complications, especially during the pandemic. This needs to be addressed as a concern/limitation in the Discussion, although it is at least addressed by interviewees on page 21, 16-30.
--

	Page 24: Regarding this quote, the authors seem to be making the assumption that this nasal spray is highly effective, cheap, and without harms. I don't have any evidence of that. They need to avoid the impression of being cheerleaders for the product: "This, alongside the identified focus on side effects and concerns about using medicines, suggests that causal learning of a treatment benefit may be difficult and this may undermine adherence. An intervention to support and promote nasal spray use will need to help people form not only strong outcome expectancies but also mental models of treatment mechanisms that allow them to conclude that the spray is working (or partially working) to counteract more pessimistic or confusing conclusions that may arise if they are primarily guided by symptom and side-effect experiences." Tables and text provide similar content and add to length. Discussion far too long. That's all I have to offer. I do research on RTI, but it is not this kind of qualitative work.
--	---

REVIEWER	Aleksandra Borek Oxford University, Nuffield Department of Primary Care Health Sciences
REVIEW RETURNED	14-Mar-2022

GENERAL COMMENTS	The article is very well-written and interesting to read. The methods are comprehensively and clearly reported, and the reporting gives confidence about the quality of the studies. The findings are also clearly reported and easy to follow and understand. The authors provided quite extensive and detailed supplementary materials, which are very helpful and interesting to read as well. The research reported in this article forms part of a larger study, including an intervention development and evaluation. It is important and helpful to publish these findings as they helped develop and refine the intervention. I believe the article could be published as it is. However, I have a couple of minor comments that the authors may consider. The authors report using thematic analysis and reference Braun and Clarke's paper from 2006. Braun and Clarke have considerably developed their ideas and suggestions around thematic analysis since that paper. For example, the methods and findings reported by the authors may fit more with what Braun and Clarke 2021 (https://doi.org/10.1080/14780887.2020.1769238) describe as a 'codebook TA'. The themes in study 2, seem a bit more like categories centred around key topics/concepts rather than themes as expressing/capturing shared meanings. For example, 'consequences of feeling protected' is a category that captures different types of perceived consequences but as such does not allow understanding what these perceived consequences were. I'd encourage the authors to consult the recent paper by Braun & Clarke 2021 and consider clarifying the type of thematic analysis approach taken. I wonder whether were there any notable differences/changes in views after participants used the spray, compared to the views expressed in think aloud interviews? Also:  • p. 7, line 16 – the word 'be' is missing ('in vitro will be less
--

	effective')  • p. 8, line 3 – the word 'be' is missing ('persistence may be difficult')
--	---

VERSION 1 – AUTHOR RESPONSE

Reviewer 1 (Prof. Mark Ebell): Comments

Comment	Response
I'm afraid that I am not a qualitative researcher and therefore can really only comment on the medical/scientific content and the presentation/writing. My review is therefore more limited than usual.	Thank you for providing your insights on the paper, and for making suggested changes for improvement.
Reviewer 1: While very well written, I found the 5200 word length to be a bit overwhelming. For example, the Introduction could be halved, easily. We know respiratory infections are common and important, don't need as much of that. We don't need an overview of prevention – just tell us about what is known about the efficacy of nasal sprays for prevention.	Thank you for your comment. Qualitative papers do tend to be longer, as methods require more detailed description and results sections include description and interpretation plus quotations. Furthermore we are reporting on two linked studies, embedded within an intervention development process, which naturally leads to a longer paper. That said, we have reduced parts of the introduction in response to your suggestion. We have also thoroughly reviewed the manuscript and made further cuts where possible.
Reviewer 1: Using a nasal spray to prevent RTI is a new concept to me – is there any evidence of efficacy, or just ongoing trials? The authors seem to make the assumption that it is highly effective, safe and cheap, and we just need to convince everyone to use it.	We recognise the point here about assuming effectiveness, safety and low cost. We have modified the manuscript to adopt a more cautious tone e.g. included more caveats about IF the product proves effective. Safety has been established for some of these products including the specific spray we developed our intervention around and are currently evaluating in our RCT, so we have added this information into the manuscript. (page 4 introduction and page 6 method) Cost of the nasal sprays that are currently on the market, including the specific product our team is investigating, is low (under £10 for a bottle that will last several months, depending on how often a person

	is in high risk/exposure situations). We have added a sentence relating to cost (page 4, method) We have also added more about the mechanism of action of these approaches and references to the evidence base (page 4, intro).
Page 7: Can you at least tell us the ingredients of the spray? I'm not interested in reading further unless I can be convinced it is likely or even possibly effective.	Unfortunately, we cannot provide the ingredients of the spray due to the ongoing masked trial. As mentioned above we have  1) provided more background information including possible mechanism of action (page 4, intro). 2) bolstered the overview of evidence of effectiveness (page 4, intro) We have included references 22-27 to provide more support for our claims regarding the spray's potential effectiveness; these are recent review articles discussing the existing evidence and suggesting nasal sprays, irrigation and related approaches are promising and worthy of research.
Page 8: Approach of using reviews from commercial sites is novel and interesting. How do you know that reviewers actually used the product, though?	We agree with this point and have added an acknowledgement to this in the limitations (page 23, discussion)
Table 3 and elsewhere (e.g. page 20, lines 6-14 and below in 'Considering use in risky situations'): I am concerned that from the comments, some users feel as if this is a "magic bullet" and they don't have to take other precautions. This could lead to an increased risk of infection and complications, especially during the pandemic. This needs to be addressed as a concern/limitation in the Discussion, although it is at least addressed by interviewees on page 21, 16-	Thank you for raising this important issue. The possibility of spray users reducing other precautionary behaviours is a concern that must be taken seriously. However, as this was a research finding, we did not include this as a limitation. Although not part of this paper, we did consider the implications of this finding in our intervention design, carefully constructing messages that supported using the spray alongside other preventative behaviours. We did not find people had formed unrealistic expectations about how effective a spray might be. We probed carefully around this issue within our study 2 interviews because it was a concern of the research team to ensure the balance between forming positive enough beliefs to motivate spray use but not to reduce caution with

30.	regard other behaviours, especially given the pandemic context. Overall, participants were excited and optimistic about trying a spray. They understood the need for further research and that it was not a ‘magic bullet’. They also wanted to try it/see for themselves how useful it may be, adding it alongside their already established preventative behaviours. None of our participants shared intentions to dramatically change their own existing precautionary behaviours, beyond what the general trend in behaviour was at the time of the study. The COVID context (summer 2020) is relevant to the quotes presented here; these participants were cautiously emerging from long, harsh lockdowns and starting to try to regain some sense of normality with work and social life; for some this was as small-scale as being able to see a limited number of family members again after being completely isolated. They tended to adopt language like ‘another layer of protection’ and talked more about emotions (feeling less anxious) rather than intending to take risks. Importantly, these insights guided the intervention development/refinement – helping us to hone our messages around potential spray benefits and the need to continue with other precautionary measures. We have added a sentence to the discussion about this: “We promoted the benefits of feeling protected, whilst explaining the importance of continuing other RI prevention behaviours” (page 22, intervention development) Process analysis embedded within our current RCT will address this concern about potential harms of spray. We are exploring (using both qualitative and quantitative methods) if and how spray use impacts upon other RTI protection behaviours such as handwashing, social distancing, face-covering. This work is ongoing but an early and tentative insight is that for some spray users, engaging with spray use seems to be drawing their attention to risky situations and potentially increasing other protective behaviours.
Page 24: Regarding this quote, the authors seem to be making the assumption that this nasal spray is highly effective, cheap, and without harms. I don’t have any evidence of that. They need to avoid the impression of being cheerleaders for the product: “This, alongside the identified focus on side effects	The quoted content has been edited in the process of reducing the word count. We feel that this comment has been addressed by adding in further information regarding safety and mechanism of action. We have also explained in the manuscript about encouraging use of the spray being a key part of the trial process in order to help to establish if they spray is effective in this context.

and concerns about using medicines, suggests that causal learning of a treatment benefit may be difficult and this may undermine adherence. An intervention to support and promote nasal spray use will need to help people form not only strong outcome expectancies but also mental models of treatment mechanisms that allow them to conclude that the spray is working (or partially working) to counteract more pessimistic or confusing conclusions that may arise if they are primarily guided by symptom and side-effect experiences.”	
Tables and text provide similar content and add to length.	We have removed the summary sentences before Table 2 (page 14, findings) as this was superfluous. Table 1 and Table 3 contain example quotations for each theme which is recommended practice in qualitative study reporting. We have followed guidelines (COREQ) to support this. Readers desiring a briefer read can still gain a sense of the findings without consulting these tables if they wish. Note that Reviewer 2 says the methods and findings reporting are comprehensive, clear and establish the reader's confidence of quality.
Discussion far too long.	We agree that the discussion could be reduced, so we have edited the discussion to make it more concise. We believe that the final discussion length is necessary to adequately explain the implications and links to theory which we hope intervention developers and behavioural scientists will find particularly useful. We have also required more words than typical to comment on the strengths and limitations, given that we report two linked studies.

Reviewer 2 (Dr. Aleksandra Borek): Comments

Comment	Response
The article is very well-written and interesting to read. The methods are comprehensively and clearly reported, and the reporting gives confidence about the quality of the studies. The findings are also clearly reported and easy to follow and understand. The authors provided quite extensive and detailed supplementary materials, which are very helpful and interesting to read as well. The research reported in this article forms part of a larger study, including an intervention development and evaluation. It is important and helpful to publish these findings as they helped develop and refine the intervention. I believe the article could be published as it is. However, I have a couple of minor comments that the authors may consider.	Thank you for your positive comments, and suggestions for improvement.
The authors report using thematic analysis and reference Braun and Clarke's paper from 2006. Braun and Clarke have considerably developed their ideas and suggestions around thematic analysis since that paper. For example, the methods and findings reported by the authors may fit more with what Braun and Clarke 2021 (https://doi.org/10.1080/14780887.2020.1769238) describe as a 'codebook TA'. The themes in study 2, seem a bit more like categories centred around key topics/concepts rather than themes as expressing/capturing shared meanings. For example, 'consequences of feeling protected' is a category that captures different types of perceived consequences but as such does not allow understanding what these perceived consequences were. I'd encourage the authors to consult the recent paper by Braun & Clarke 2021 and consider clarifying the type of thematic analysis approach taken.	Thank you. We agree that our approach is not completely aligned with Braun and Clarke's more recent discussions of thematic analysis. We have removed the reference to Braun and Clarke for this reason and have instead focused on describing our actual analytic steps.
I wonder whether were there any notable differences/changes in views after participants used the spray, compared to the views expressed in think aloud interviews?	With the exception of 2 people, it was different participants in think-aloud and post-intervention use interviews. Our intervention was refined in between study 2a and 2b and therefore people did respond a little differently e.g. disgust/hygiene issue was more prominent in the think-alouds and seemed to be improved when we changed our wording from something like 'think of it like washing your hands' to 'think of it like using hand gel'. But there were no major differences to report in terms of overall themes.
p. 7, line 16 – the word 'be' is missing ('in vitro	This sentence has now been removed in response to comments from Reviewer 1 about

will be less effective')	reducing the length of the introduction?.
p. 8, line 3 – the word 'be' is missing ('persistence may be difficult')	Thank you. This change has been made.

VERSION 2 – REVIEW

REVIEWER	Aleksandra Borek Oxford University, Nuffield Department of Primary Care Health Sciences
REVIEW RETURNED	01-Jun-2022

GENERAL COMMENTS	I have no further comments.
-----------------------------